# Enhancing Mild Cognitive Impairment Auxiliary Identification Through Multimodal Cognitive Assessment with Eye Tracking and Convolutional Neural Network Analysis

**DOI:** 10.3390/biomedicines13030738

**Published:** 2025-03-18

**Authors:** Na Li, Ziming Wang, Wen Ren, Hong Zheng, Shuai Liu, Yi Zhou, Kang Ju, Zhongting Chen

**Affiliations:** 1Shanghai Changning Mental Health Center, Affiliated Mental Health Center of East China Normal University, Shanghai 200335, China; nli@psy.ecnu.edu.cn (N.L.);; 2Psychology and Cognitive Science, East China Normal University, Shanghai 200062, Chinawendyr1@163.com (W.R.);

**Keywords:** eye movements, MCI, CNN analysis, auxiliary identification

## Abstract

**Background**: Mild Cognitive Impairment (MCI) is a critical transitional phase between normal aging and dementia, and early detection is essential to mitigate cognitive decline. Traditional cognitive assessment tools, such as the Mini-Mental State Examination (MMSE) and the Montreal Cognitive Assessment (MoCA), exhibit limitations in feasibility, which potentially and partially affects results for early-stage MCI detection. This study developed and tested a supportive cognitive assessment system for MCI auxiliary identification, leveraging eye-tracking features and convolutional neural network (CNN) analysis. **Methods**: The system employed eye-tracking technology in conjunction with machine learning to build a multimodal auxiliary identification model. Four eye movement tasks and two cognitive tests were administered to 128 participants (40 MCI patients, 57 elderly controls, 31 young adults as reference). We extracted 31 eye movement and 8 behavioral features to assess their contributions to classification accuracy using CNN analysis. Eye movement features only, behavioral features only, and combined features models were developed and tested respectively, to find out the most effective approach for MCI auxiliary identification. **Results**: Overall, the combined features model achieved a higher discrimination accuracy than models with single feature sets alone. Specifically, the model’s ability to differentiate MCI from healthy individuals, including young adults, reached an average accuracy of 74.62%. For distinguishing MCI from elderly controls, the model’s accuracy averaged 66.50%. **Conclusions**: Results show that a multimodal model significantly outperforms single-feature models in identifying MCI, highlighting the potential of eye-tracking for early detection. These findings suggest that integrating multimodal data can enhance the effectiveness of MCI auxiliary identification, providing a novel potential pathway for community-based early detection efforts.

## 1. Introduction

Mild Cognitive Impairment (MCI) is a critical transitional phase between normal age-related cognitive decline and the onset of dementia [1]. Specifically, MCI is characterized by a noticeable deterioration in cognitive functions, including memory, executive functions, language, and visuospatial skills, while daily living capabilities remain largely intact [2]. According to Jia et al. (2020) [3], the prevalence of MCI in China has reached 15.5%, affecting approximately 38.77 million people. Given this increasing prevalence, early and accurate detection of MCI is essential for timely intervention to mitigate further cognitive decline.

Traditional cognitive assessments are the most used methods in MCI early detection, such as the Mini-Mental State Examination (MMSE) and the Montreal Cognitive Assessment (MoCA), although they have some inevitable limitations in practice [4,5,6]. For instance, the comprehensive nature of the MoCA necessitates clinical expertise for administration, which hinders broader detection efforts. Higher-educated individuals typically achieve better scores [7], which can lead to misinterpretation of cognitive impairment, especially among illiterate individuals who tend to score significantly lower compared to literate individuals [8]. Meanwhile, in the case of the MMSE, despite being more accessible for individual use, its sensitivity and stability are compromised due to educational bias, affecting its diagnostic accuracy [3,9]. It exhibits a ceiling effect, with an optimal cutoff point of 29/30 resulting in only 54% correct diagnoses of cognitive impairment, indicating its limited sensitivity for detecting mild cognitive impairment (MCI) [10]. Moreover, both the MMSE and MoCA rely on subjective scoring, particularly for open-ended questions and tasks that require interpretation (e.g., drawing or construction tasks). This can lead to variability in scores depending on the scorer’s judgment, which may affect the reliability of the results [11]. Therefore, there is an urgent need for automated measurement methods that can efficiently and accurately assist in the identification of MCI, ideally being non-invasive, easy to administer, and capable of overcoming the limitations of current tools.

In this context, eye-tracking technology has emerged as a promising non-invasive alternative for identifying and monitoring cognitive decline among various population groups [12]. Recent research has underscored the effectiveness of eye tracking in distinguishing individuals with Alzheimer’s disease (AD) from MCI patients and healthy controls, utilizing indices such as reaction time and error rates from an anti-saccade task [13]. Another study has demonstrated a novel 3 min eye-tracking test’s efficacy in assessing cognitive function among normal controls, MCI, and AD subjects, particularly focusing on memory and deductive reasoning tasks [14]. Prior studies have validated the effectiveness of eye movement tracking techniques in differentiating cognitive impairments, highlighting its utility in the early identification and monitoring of cognitive decline [15,16].

In addition, the introduction of machine learning, particularly into the realm of cognitive impairments, enhances diagnostic precision and reliability of using biomarkers for early identification and treatment strategies [17]. For example, Convolutional Neural Network (CNN) models have demonstrated high accuracy in classifying Alzheimer’s disease and predicting the transition from MCI to Alzheimer’s dementia [18,19]. Combining eye-tracking data with machine learning has shown significant potential in improving diagnostic accuracy, with algorithms like Support Vector Machines (SVM) and Bayesian classifiers achieving over 80% accuracy in assessing cognitive disorders [20]. Likewise, Lin et al. (2024) [21] employed the Light Gradient Boosting Machine (LightGBM) to combine gait and eye movement features for accurately identifying cognitive impairments. Overall, advanced machine learning classifiers, including those that were trained with eye movement data, are a potential approach to distinguishing MCI and early Alzheimer’s with high precision [11,22].

In light of these findings, this study aims to develop a cognitive status assessment system based on eye-tracking features and deep neural network analysis using CNN. We further analyze the contribution of eye-tracking and behavioral data to classification accuracy. Our MCI dataset comprises 128 valid participants, including 57 elderly controls, 40 MCI patients, and 31 young adults. Ultimately, this research, building upon previous studies, validates the feasibility of using eye-tracking features for auxiliary identification of cognitive impairments and contributes to the development of more accessible community-based auxiliary identification systems, holding significant implications for future researchers in constructing broader identification models [23].

## 2. Materials and Methods

### 2.1. Participants

One hundred and forty-seven participants in total volunteered for the present study, comprising forty-seven MCI participants (range: 60–87 years, mean: 72.55 years, SD: 6.72 years), sixty-nine healthy control participants (range: 56–87 years, mean age: 69.7 years, SD: 7.88 years), and thirty-one healthy young participants (range: 17–27 years, mean: 21.71 years, SD: 2.60 years) as a reference group. Visual acuity was measured in all participants to ensure normal or corrected-to-normal vision. Participants in the control group and the young group had no history of neurological diseases, psychiatric disorders, or musculoskeletal dysfunctions. All participants and their guardians signed informed consent forms. This study was approved by the Ethics Committees of the Shanghai Changning Mental Health Center and East China Normal University, and it conforms to ethical standards for human research.

Participants in the healthy young control group were recruited from East China Normal University and did not undergo MCI diagnostic evaluations. MCI patients and healthy elderly participants were recruited through a collaboration between the Shanghai Changning Mental Health Center and local community committees. The MCI patients met the diagnostic criteria proposed by Petersen et al. (1999) [24]. The diagnosing process was as follows: For those with potential MCI risk, a standardized evaluation was conducted, including clinical symptoms of cognitive impairment, assessment of daily living abilities, and the severity of cognitive impairment. Specifically, (i) the clinical symptoms of cognitive impairment were confirmed by the patient, an informant, or an experienced clinician; (ii) the Activities of Daily Living (ADL, Barthel Test) scale was used to assess daily living abilities; and (iii) due to logistical constraints, the assessment of cognitive impairment severity was divided into two batches. The first batch used the Mini-Mental State Examination (MMSE) scale to assess 62 individuals, with 3 identified as MCI patients. The second batch used the Montreal Cognitive Assessment (MoCA) scale to assess 54 individuals, with 44 identified as MCI patients. In total, 47 MCI patients were selected.

### 2.2. Apparatus

The participant was seated comfortably in a dim room facing an LED monitor (Lenovo, Beijing, China, D27-30, 1920 × 1080 pixels, 27-in., 60 Hz) positioned in the frontal plane 50 cm from the participant’s eyes. Head movements were restrained by a chin rest so that the eyes were directed toward the center of the screen. The movements of the left eye were recorded at a sampling rate of 60 Hz using a Gazepoint3 eye tracker (Gazepoint Vancouver, BC, Canada). The experiment was programmed in MATLAB (version: R2018b) with the Psychtoolbox package [25,26,27].

### 2.3. Task and Procedure

#### 2.3.1. Eye Movement Tasks

Four visual tasks were designed and performed in separate sessions: a pro- and anti-saccades task [28,29], a smooth pursuit task [30], a memory-guided saccades task [31,32,33], and a predictive saccades task [34,35] (see Figure 1).

Pro- and anti-saccades task: A drift correction procedure was performed before each trial. Participants were asked to look at a fixation dot at the center of a screen on a gray background and press the space bar to start. A black fixation cross (+; 1° × 1° of visual angle) was initially presented at the screen center, and four black circles (diameter: 0.5° of visual angle) were displayed at ±9.7° horizontally and vertically away from the screen center at the same time. Firstly, participants were required to fixate the cross (which lasted between 1000 and 1500 ms). Then, in each trial, one of the black circles turned blue or red randomly. In the pro-saccade task, participants were required to look at the target (which is the blue circle). In the anti-saccade task, participants were required to look at the opposite position to the target (which is the red circle). For trials that timed out (participant failed to look to the color-changed circle 1000 ms after target onset) and for trials in which participants made an initial saccade to the wrong position, participants received visual (“wrong”) feedback which lasted 200 ms. Each participant first finished 16 practice trials including 2 times for each direction for pro-and anti-saccade conditions and then started the formal experiment, which was performed 8 times during 2 blocks for pro- and anti-saccade tasks each, with a total number of 64 trials (see Figure 1A).

Smooth pursuit task: Participants were asked to gaze at a red fixation circle (diameter: 0.5° of visual angle) at the center of the screen on a gray background. The trial started when the fixation circle turned blue after participants’ gaze exceeded 1200 ms. Participants were instructed to track along with the blue target, which moved in a horizontal or vertical direction. The total amplitude of the target movement was 20° of visual angle (10° of visual angle either side of the center). The frequency of the sinusoidal target oscillation was set at 0.25 Hz and 0.4 Hz, respectively. Each participant first finished 2 practice trials (one horizontal and one vertical), each of which lasted 10 s, and then started the formal experiment, which was performed 2 times for each of the two frequencies and directions, with a total number of 8 trials (see Figure 1B).

Memory-guided saccades task: A drift correction procedure was performed before each trial. Participants were asked to look at a fixation dot at the center of a screen on a gray background and press the space bar to start. A black fixation cross (+; 1° × 1° of visual angle) was initially presented at the screen center. Firstly, participants were required to fixate the central cross (which lasted between 1000 and 1500 ms). Then, on each trial, a peripheral target (diameter: 0.5° of visual angle) was presented for 100 ms at 5°, 10°, or 15° horizontally from the center of the screen, giving a total of 6 target locations. After a variable, 4500 to 5000 ms delay, the cross disappeared. Until this point, participants were instructed to gaze at the central cross. At this point, participants were asked to make a memory-guided saccade (looking at the remembered location of the peripheral target) while the screen remained blank for 1000 ms. Each participant first finished 12 practice trials including 2 times for each direction and then started the formal experiment, which was performed 10 times during 2 blocks for each direction, with a total number of 60 trials (see Figure 1C).

Predictive Saccade task: Participants were asked to gaze at a red fixation circle (diameter: 0.5° of visual angle; which lasted between 1000 and 1500 ms) at the center of the screen on a gray background. In each trial, the blue target circle (diameter: 0.5° of visual angle) alternated between the two fixed locations (10° of visual angle either side of the screen center) on the horizontal plane for a total of 12 target steps. A square-wave target alternated consistently at one of the five target rates (0.66, 0.8, 1, 1.33, and 2 Hz). These five target rates correspond to the following ISIs (interstimulus intervals): 1500, 1250, 1000, 750, and 500 ms, respectively. Each participant first finished 2 practice trials for two frequencies (0.66 and 2 Hz) and then started the formal experiment, which was performed 5 times during 5 blocks for each frequency, with a total number of 25 trials (see Figure 1D).

#### 2.3.2. Tests of Cognitive Capabilities

Participants’ cognitive capabilities were measured in two domains, inhibitory control function, and working memory.

Stroop Task: Participants first saw a cross fixation point. Subsequently, participants were presented with a color–word combination and were required to quickly and accurately judge the color of the word and press the corresponding color key. The task included three words (red, green, blue) and three presentation colors (red, green, blue). The practice phase consisted of 9 trials, with each level presented once; the formal experiment consisted of 90 trials, with each stimulus level presented 10 times, and the order of stimulus presentation was random. During the practice, feedback on the correctness of the judgment was provided to help participants understand the operation of the experiment, while there was no feedback in the formal experiment. The task analyzed the reaction times of participants under conditions where the color and the word were consistent and inconsistent. The results of the Stroop task calculated accuracy, reaction time, accuracy under the word–color consistent condition, reaction time under the word–color consistent condition, accuracy under the word–color inconsistent condition, and reaction time under the word–color inconsistent condition.

Working Memory Tasks: Participants performed forward/backward visuospatial tasks and forward/backward digit span tasks.

In the forward/backward visuospatial task, participants saw a 16-gallery grid on the screen. During each trial, a fixed number of apples appeared in this grid subsequently and randomly. Participants needed to remember the order and location of each apple’s appearance. After the items were presented, a blank background appeared for 2000 ms. Then, a blank 16-gallery grid was presented, and participants needed to light the apples at the corresponding locations sequentially on the computer, either in order (forward) or in reverse order (backward). The task’s difficulty increased with the length of the items to be recalled. Each participant first finished 3 practice trials for 2 items and then started the formal experiment, which began with 3 items. If these were not recalled correctly, “error” feedback was given; if the answers were incorrect for 3 consecutive times, the test was automatically terminated. The task was scored by awarding one point for each correct trial.

In the forward/backward digit span tasks, participants heard a series of numbers and needed to memorize the order in which the numbers appeared. After the numbers were presented, a blank background appeared for 2000 ms. Then, participants needed to recall the numbers they heard and enter them in the correct sequence in a blank input box on the computer, either in order (forward) or in reverse order (backward). The task’s difficulty increased with the length of numbers to be recalled. Each participant first finished 3 practice trials for 2 numbers and then started the formal experiment, which began with 3 numbers. If they were not answered correctly, “error” feedback was given; if the answers were incorrect for 3 consecutive times, the test was automatically terminated. The task was scored by awarding one point for each correct trial.

In consideration of the potential impact of testing duration and fatigue on the results in elderly patients, we implemented several measures to ensure accurate and reliable data collection. During tasks such as the smooth pursuit task, rest periods were incorporated to prevent fatigue, and the task would only commence after the participant had fixated on the target for a certain duration. Rest periods were also arranged between different blocks of the task, with the duration adjusted according to the participant’s condition and well-being. Both behavioral and eye movement tests included longer rest periods to ensure participants were not fatigued at the start of the tests. The overall duration of the tests varied by individual, with eye movement tests taking approximately 30 min and behavioral tests around 20 min. Additionally, the testing time was individualized for each participant to ensure their comfort and to minimize the impact of fatigue on the results.

### 2.4. CNN Analysis

In this study, we use the PyTorch package (version: 2.2.0) in Python 3 to build Convolutional Neural Networks (CNNs) for classification. Eye-tracking and behavioral features were not provided to the CNN in video format but were extracted as scalar features from four eye-movement tasks (pro- and anti-saccades task, smooth pursuit task, memory-guided saccades task, and predictive saccade task) and two cognitive tasks (Stroop Task and Working Memory Tasks) using Matlab. In this study, the input data comprised two sets, namely, eye-movement-related data and behavioral data, comprising 31 eye-movement features and 8 behavioral features. We concatenated the eye-tracking features and behavioral features into a single feature vector, meaning that each participant had 39 features that were fed into the model. The features from specific tasks and measurements are listed in Table 1. For details on definitions of input features, please see Appendix A.

The model consists of two one-dimensional convolutional layers and two one-dimensional pooling layers stacked alternately, followed by three fully connected layers for the output. The kernel size of each convolutional layer is 2, with a stride of 1. The number of output channels for the first convolutional layer is 16, while the number of output channels for the second convolutional layer is 32. This is carried out to enhance the model’s feature representation capability and to capture higher-level abstract features from the data. To ensure that the input and output features have the same dimensions during each convolution operation, padding was applied. We choose max pooling to reduce computational load. After the convolutional layers and pooling layers, a flattening layer was added to convert the multi-dimensional feature maps into a one-dimensional vector, making it suitable for subsequent fully connected layers.

The first fully connected layer outputs 4096 features, and the expanded dimensions help the network capture more complex data characteristics. The second layer outputs 256 features, which simplifies the rich feature information from the previous layer and focuses on more representative features. The third fully connected layer is the final output layer, with the number of output features corresponding to the number of categories (2 for binary classification, 3 for triple classification), representing the final classification result. We incorporate Dropout layers in the network, which are regularization techniques that prevent overfitting by randomly dropping neurons during training. Considering the relatively small sample size and the data volume of the current study, larger kernels and more convolutional layers did not effectively improve classification accuracy, so we adjusted the number of convolutional layers, kernel size, stride, number of hidden layers, and batch size to adapt the data size of the current study.

In convolutional neural networks, if there are no activation functions in the CNN, the relationship between neuron outputs and inputs remains linear. To enhance the network’s ability to express nonlinearities, activation functions should be added between different layers, thus imparting nonlinear characteristics to the data and enhancing model stability. In this paper, the ReLU function (f(x) = max(0,x)) was used as the activation function. Compared to other activation functions such as Sigmoid, the ReLU function can effectively avoid the problem of vanishing gradients and set the output of some neurons to zero during computation, alleviating overfitting issues in the small sample dataset used in this paper.

The essence of CNN lies in computing the loss function and continuously updating model weights by monitoring the decrease in the loss function to optimize the model. For the choice of loss function, we adopted the Cross Entropy Loss. Cross Entropy Loss can be used in binary and multiclass tasks. It calculates the cross-entropy loss between the model output and the true labels, serving as the objective function for model optimization. The smaller the value of cross-entropy, the better the model prediction performance.

We selected the Adam optimizer with an initial learning rate of 0.001. The first-order momentum parameter β1 and the second-order momentum parameter β2 were set to their default values (β1 = 0.9, β2 = 0.999). The Adam optimizer is an adaptive optimization algorithm that adjusts the learning rate based on historical gradient information, allowing the model to converge quickly in the early stages of training and to find the minimum of the loss function faster in the later stages. Additionally, the Adam optimizer can adjust momentum parameters to prevent falling into local minima. To further optimize the model in the later stages of training, we used the StepLR learning rate scheduler. This scheduler reduces the learning rate by 10% after every 10 epochs, helping to prevent oscillations in the later stages of training that can be caused by large parameter updates.

For the binary and ternary classification tasks, the ratio of the training set to the test set was set to 8:2. For each classification task, 10 independent classification results were independently computed, with outputs of classification accuracy, loss function, participant IDs with classification errors, and labels of misclassification. While maintaining the same ratio, different divisions of training and test sets were employed in each classification to confirm model generalization. Subsequently, the average and variance of the classification accuracy over 10 results were calculated as the final output results.

## 3. Results

Eye movement data and behavioral data were collected from a total of 147 participants. After excluding participants who did not complete all tasks and those with recording issues, 128 valid participants remained, comprising 40 MCI patients (range: 60–87 years, mean: 72.60 years, SD: 6.22 years, 30 women), 57 healthy elderly adults (range: 56–87 years, mean: 69.39 years, SD: 5.10 years, 34 women), and 31 young adults (range: 17–27 years, mean: 21.71 years, SD: 2.60 years, 20 women).

### 3.1. Differences in Eye Movement and Behavioral Features Between Groups

To further investigate the characteristics of eye movement performance across groups, we extracted key features from a total of 31 eye movement parameters. Initially, we excluded ten indicators that did not show significant differences between young and elderly adults. From the remaining 21 features, we selected 11 that effectively reflect the overall characteristics of the task. These 11 features demonstrated significant differences between the young and elderly groups. These features reflect the accuracy, reaction speed, and stability in eye movement tasks (see Figure 2 and Appendix A).

Analysis results showed that the young group performed significantly better than the elderly group (including the control and MCI groups) across these 11 features, |*ts*| > 2.06, *ps* < 0.042, indicating that young participants outperformed the elderly in accuracy, reaction speed, and stability. Within the elderly groups, the control group demonstrated no significant difference from the MCI group (|*ts* (95)| < 1.49, *ps* > 0.14). Data distribution of eye movement variables under four tasks were derived to illustrate differences among the three groups (see Figure 2 and Appendix A). For the working memory tasks, the young group performed the best as well, followed by the elderly control group, with the MCI group performing the worst. The young group performed significantly better than the elderly group (including the control and MCI groups), |*ts* (75)| > 11.66, *ps* < 0.001. The young group also performed significantly better than the elderly group on the response time but not the accuracy of the Stroop test. Within the elderly group, the MCI group performed significantly worse than the control group on the backward digit span task (*t* (94.25) = 3.42, *p* = 0.001), with no significant differences observed in other cognitive tasks (|*ts*| < 1.97, *ps* > 0.052) (see Appendix A).

### 3.2. Relationship Between Eye Movement Variables and Behavioral Features

Pearson correlation analysis between the eye movement features and behavioral features was conducted to investigate the relationships between eye movement and behavioral features. The accuracy and variance features observed from the eye movement tasks demonstrated positive correlations with the four working memory tasks, whereas there were significantly negative correlations between those features and the response time of the Stroop test for inhibitory control function, with the exception of variance horizontal at 0.25 Hz and 0.4 Hz in the smooth pursuit task (*ps* > 0.05). For instance, the pro-saccades accuracy was positively correlated with the forward visuospatial task (*r* = 0.45, *p* < 0.01) and negatively with the consistent response time of the Stroop test (*r* = −0.39, *p* < 0.01). Conversely, the latency, gain, and delay features in the eye movement tasks were negatively correlated with the four working memory tasks, but positive with the response time of the Stroop test. The accuracy of the Stroop test had non-significant correlations with the eye movement features (*ps* > 0.05). The fixation dispersion in pro- and anti-saccades tasks and smooth pursuit task was significantly correlated with some working memory tasks. For details of the correlation analysis results, please see Appendix A.

### 3.3. Power of Discriminating MCI from Healthy Individuals Including Young Adults

In the current study, seven feature sets (i.e., eye movement features only, behavioral features only, combined features, combined features without pro- and anti-saccades task, combined features without smooth pursuit task, combined features without memory-guided task and combined features without prediction saccades task) were used to train a CNN for MCI detection (see Table 2).

The model trained with the combined features achieved an average discrimination accuracy of 74.62% using CNN for feature extraction as a classifier. The model based on behavioral features only (accuracy: 72.31%) outperformed the model based on eye movement features only (accuracy: 69.62%) which yielded the lowest accuracy. Notable, the accuracy of the model dropped to 68.85% when the pro- and anti-saccades features were excluded, and it increased to 78.48% when the prediction saccade features increased. The other two models that incorporated eye movement data achieved accuracies of 70% and above.

The effect of combining features in the model on MCI participants’ identification was the best, with a hit rate of 60.77% and it was also the best model for distinguishing between elderly control and young individuals, with a hit rate of 73.08% and 100%, respectively. The next best model, which used only behavioral features, had hit rates of 67.84% for elderly control individuals and 57.50% for MCI participants. The eye movement features-only model could distinguish the elderly control individuals (hit rate: 66.60%) better than the MCI participants, and the latter case was almost at the random chance level (hit rate: 51.63%). All models could effectively distinguish young participants, with the lowest achieved hit rate being 94.90%. Removing pro- and anti-saccades features reduced the model’s hit rate from 60.77% to 51.50%, indicating the critical role of these features in identifying individuals with MCI. However, when smooth pursuit, memory-guided saccade, and predictive saccade features were excluded, the hit rates for correctly distinguishing MCI individuals all remained above 58%, and among those models removing predictive saccade features even increased the model’s hit rate to 62.64%.

### 3.4. Power of Discriminating MCI from the Healthy Elderly Individuals

For the binary classification, we used CNN analysis to discriminate MCI participants from healthy elderly individuals. As with the triple classification, the model utilizing combined features demonstrated the highest accuracy (66.50% for MCI detection). However, unlike the findings from the triple classification task, the model based solely on eye movement features outperformed the behavioral features-only model, with accuracies of 61% and 58.50%, respectively (see Table 3). The model using only eye movement features also achieved a better result (AUC: 0.61) than the behavioral features-only model (AUC: 0.51). Moreover, the combined features model achieved the best AUC at 0.64 (see Figure 3). The transition from triple to binary classification did not significantly disrupt the stability of the models, though the accuracy of the behavioral features-only model decreased markedly in the absence of young adults from the dataset. Notably, all four of the models that incorporated eye movement data achieved accuracies of 61.00% and above.

Among those models, the combined features model was relatively effective in distinguishing MCI participants from the normal population with similar age and gender characteristics, achieving a hit rate of 59.76% for MCI patients and 70.02% for healthy elderly individuals. The model that used only behavioral features yielded a hit rate of 52.79% for correctly identifying MCI patients, while the eye movement features-only model had the lowest hit rate at 49.31%. Excluding the predictive saccade task from the combined model increased the hit rate to 59.88%, and almost had the same hit rate as the combined features model. Other sets of models incorporating eye movement data achieved hit rates between 52.77% to 56.97% in identifying MCI patients.

### 3.5. Comparison of CNN with Other Models

To further validate the advantages of CNN in distinguishing MCI, we compare the classification performance of CNN with that of Support Vector Machine (SVM) and Random Forest. First, regarding SVM, it showed a weaker ability to distinguish MCI patients from the control group of elderly individuals, with an average accuracy of 53.00%. The accuracy for distinguishing MCI was only 51.15%, which is not much different from random chance (50%). However, SVM was able to distinguish MCI patients from young individuals, with an accuracy of 99.44%. The classification performance of Random Forest was similar to that of SVM. In the binary classification task, Random Forest’s average accuracy was 53.00%, with the accuracy for distinguishing MCI being low (47.80%), below random probability. In the triple classification task, Random Forest’s average accuracy was 63.46%, with an accuracy of 100% for distinguishing young individuals, but only 43.6% for distinguishing MCI patients. Both SVM and Random Forest significantly underperformed compared to CNN, which achieved an average accuracy of 66.50% in binary classification and 74.62% in triple classification.

To explore the importance of the CNN convolutional operation in distinguishing MCI patients, we compared the classification performance of CNN and FCNN (Fully Connected Neural Network). In the binary classification task, the average accuracy of FCNN was 57.50%, with an MCI classification accuracy of 51.94%. In the triple classification task, the average accuracy of FCNN was 67.69%, with a classification accuracy of 100% for young individuals and 47.09% for MCI patients.

## 4. Discussion

Our study applies a multimodal analysis combining eye movement and behavioral features to identify MCI. By extracting 31 eye movement and 8 behavioral features, we constructed a CNN-based machine learning model to compare identification performance among single behavioral features, single eye movement features, and their fusion. Our findings indicate that the fusion model effectively distinguishes MCI patients, particularly from healthy individuals, underscoring its clinical value for early MCI diagnosis. This approach provides a valuable reference for AD prevention and complements prior research focused predominantly on diagnosing AD, contributing to a more comprehensive understanding of cognitive decline disorders.

Our results align with previous cognitive impairment research [36,37,38,39], confirming that MCI patients exhibit deficits in memory, particularly in spatial and digit recall, as well as poorer working memory performance compared to healthy elderly individuals, evidenced by lower backward digit span scores. Eye movement features, recognized as biomarkers for early AD stages and cognitive impairment assessment, have shown high accuracy in eye-tracking-based cognitive tasks [40,41]. However, our study reveals that while eye movement features can distinguish between young adults and older adults, they are less effective in differentiating MCI patients from elderly controls, highlighting diagnostic challenges specific to MCI.

Prior studies have focused mainly on MCI progression to severe cognitive impairments like AD, with visual attention models achieving high dementia prediction accuracy [42]. Eye-tracking data for cognitive impairments have demonstrated a strong link between eye movement and cognitive functions [43]. For example, dual-task eye-tracking and gait analysis achieved an AUC of 0.866–0.893 in distinguishing MCI from dementia [21]. Conversely, identifying MCI from the general population remains challenging. In our study, using only eye movement features yielded an accuracy of 61% and an AUC of 0.61; adding behavioral features improved the accuracy and AUC to 66.5% and 0.64, respectively. Studies combining gait and eye movement tasks reported similar AUCs (0.713–0.742) for distinguishing MCI from cognitively normal (CN) individuals [21]. Despite a trend toward improved accuracy and specificity with added features, overall model performance remained modest. In addition to the findings presented, our study suggests that the eye-tracking data captured in this research are primarily reflective of cognitive inhibition, a key component of executive function under high cognitive load. While saccadic movements have been hypothesized to relate to working memory through mechanisms that are still not fully understood, our results indicate that these movements are more closely associated with cognitive inhibition. This finding is consistent with studies showing the importance of cognitive inhibition in maintaining focus and filtering out irrelevant information, particularly in tasks requiring high cognitive load. Future research should explore the potential links between saccadic movements, cognitive inhibition, and direct cognitive load to further elucidate these mechanisms.

Our findings reveal that classification using only eye movement features performed poorly in distinguishing MCI patients from healthy elderly individuals, with an accuracy of 61.0%—though higher than their performance (Accuracy: 58.5%) using behavioral features. Conversely, using behavioral features only demonstrated a higher accuracy of 72.31% than eye movement features only, which had an accuracy of 69.62%, in distinguishing young groups from older groups (MCI and elderly controls). This suggested that traditional assessments are more effective in differentiating between age groups due to their comprehensive evaluation of age-related cognitive declines [44]. For example, motor limitations among older adults may impede their performance in behavioral tasks, and the diverse educational backgrounds of the elderly are frequently neglected in conventional assessments, which may introduce bias [45,46]. Meanwhile, this sensitivity of eye movement metrics to subtle cognitive changes, especially in MCI patients, is supported by evidence showing that tasks like the anti-saccades paradigm can effectively differentiate MCI patients from healthy controls [13]. Consequently, eye movement tasks are more effective in identifying cognitive impairments associated with MCI, making them a valuable tool in the assessment of cognitive health in the elderly [13,47]. We observed that the combination of eye movement metrics, including pro- and anti-saccades, smooth pursuit, memory-guided saccades, and prediction saccades, achieved the highest accuracy of 66.5% in distinguishing individuals with MCI from healthy controls, indicating that a single metric may not fully capture an individual’s cognitive state due to the diverse aspects of cognitive function they reflect. Compared to traditional self-reporting methods, eye-tracking technology reduces subjective bias, enhancing the objectivity of detection. Furthermore, the application of machine learning algorithms enables the automatic identification of patterns related to MCI from large datasets. This offers a more convenient, objective, and automated means for cognitive assessment, facilitating preliminary auxiliary identification and evaluation of cognitive function.

In this study, by comparing the performance of the CNN model with other models (such as FCNN) in classification tasks, we find that the CNN model has a significant advantage in distinguishing MCI patients. Specifically, the classification accuracy of the CNN model (with an average accuracy of 66.50% in binary classification and 74.62% in triple classification) is significantly higher than that of the FCNN model (with an average accuracy of 57.50% in binary classification and 67.69% in triple classification). This suggests that the convolutional operation plays an important role in distinguishing MCI from the elderly control group. Although CNN is typically used for processing high-dimensional data (such as images or videos), the results of this study show that even when the input data consist of scalar features, the discriminatory power of CNN remains significantly stronger than that of other models. This phenomenon may be attributed to the fact that the convolutional layers of CNN can extract high-dimensional features from the data, and these features, compared to individual eye-tracking features (such as fixation dispersion), are more indicative of the participants’ cognitive states. Specifically, the convolutional operation performs local conjunctions of features in the input data through sliding convolution kernels, thus generating higher-dimensional features [48]. For example, in the smooth pursuit task, fixation dispersion and saccade compensation, when considered separately, may not provide strong discriminatory power, but their combination may be crucial for assessing cognitive state. The convolutional operation can learn the local patterns among these features, thereby extracting feature combinations that are more effective in distinguishing MCI patients.

Previous research has highlighted the potential of microsaccades—fine involuntary eye movements—as a valuable metric for assessing cognitive function. Microsaccades are closely linked to attention and cognitive processing [49] and have been shown to be highly sensitive to cognitive load in tasks requiring rapid shifts of attention [50]. Additionally, they have been identified as potential indicators of neurodegenerative changes in conditions such as Alzheimer’s disease and MCI [51]. Incorporating microsaccade metrics into our current framework of eye-tracking features may improve the accuracy of early detection and provide deeper insights into the underlying mechanisms of cognitive decline. Future studies should consider the integration of microsaccade metrics to further explore their potential in enhancing diagnostic models for MCI.

Behavioral features were included to improve assessment accuracy, in recognition of the complexity of cognitive impairments. The lower accuracy observed in our study may be due to a small sample size, a common limitation in precision medicine. Nevertheless, our findings demonstrate the value of multi-dimensional parameters for MCI early detection. The heterogeneity of MCI may also explain the limited model accuracy, suggesting that a single approach may not be effective. For example, amnestic MCI (aMCI) patients, exhibit eye movement abnormalities in memory-related tasks that are not typically observed in non-amnestic MCI (naMCI) patients, reflecting the distinct memory impairments characteristic of each subtype. Considering MCI subtypes has been shown to enhance diagnostic accuracy. Our model, as an auxiliary identification tool, aiming to capture this complexity, incorporated a variety of eye movement tasks and behavioral measures to improve sensitivity. Future research on specific eye movement biomarkers and neuropsychological criteria that differentiate MCI subtypes could further aid in predicting dementia progression. A deeper understanding of MCI subtypes could facilitate targeted prevention strategies and support more effective testing of future therapeutic interventions [52,53,54,55].

## 5. Conclusions

Overall, this study validates the effectiveness of integrating eye movement and behavioral features for multimodal MCI auxiliary identification. Compared to previous studies, this research further demonstrates the advantages of multimodal features in capturing the complex cognitive impairment patterns in MCI, especially providing a more comprehensive perspective for MCI auxiliary identification. The support of eye-tracking technology also opens the possibility of examining the underlying neuronal connections, surpassing the limitations of traditional self-reported methodologies [56]. However, this study has its limitations. First, the sample size was relatively small, particularly for MCI patients, which may affect the model’s stability and generalizability. Second, some eye movement tasks may not fully capture the complexity of MCI, particularly in the predictive saccade task, where the effectiveness of MCI subtypes’ identification is limited. Future research could improve in the following areas: (1) expanding the sample size, particularly through multi-center data collection, to enhance the model’s generalizability and robustness; (2) exploring the integration of more physiological and behavioral signals, such as EEG and skin conductance response, to further improve the accuracy of early MCI identification; and (3) designing personalized cognitive tasks for different MCI subtypes to enhance the model’s sensitivity to subtype differences, to provide more practical auxiliary identification tools for clinical diagnosis and intervention.

## Figures and Tables

**Figure 1 biomedicines-13-00738-f001:**
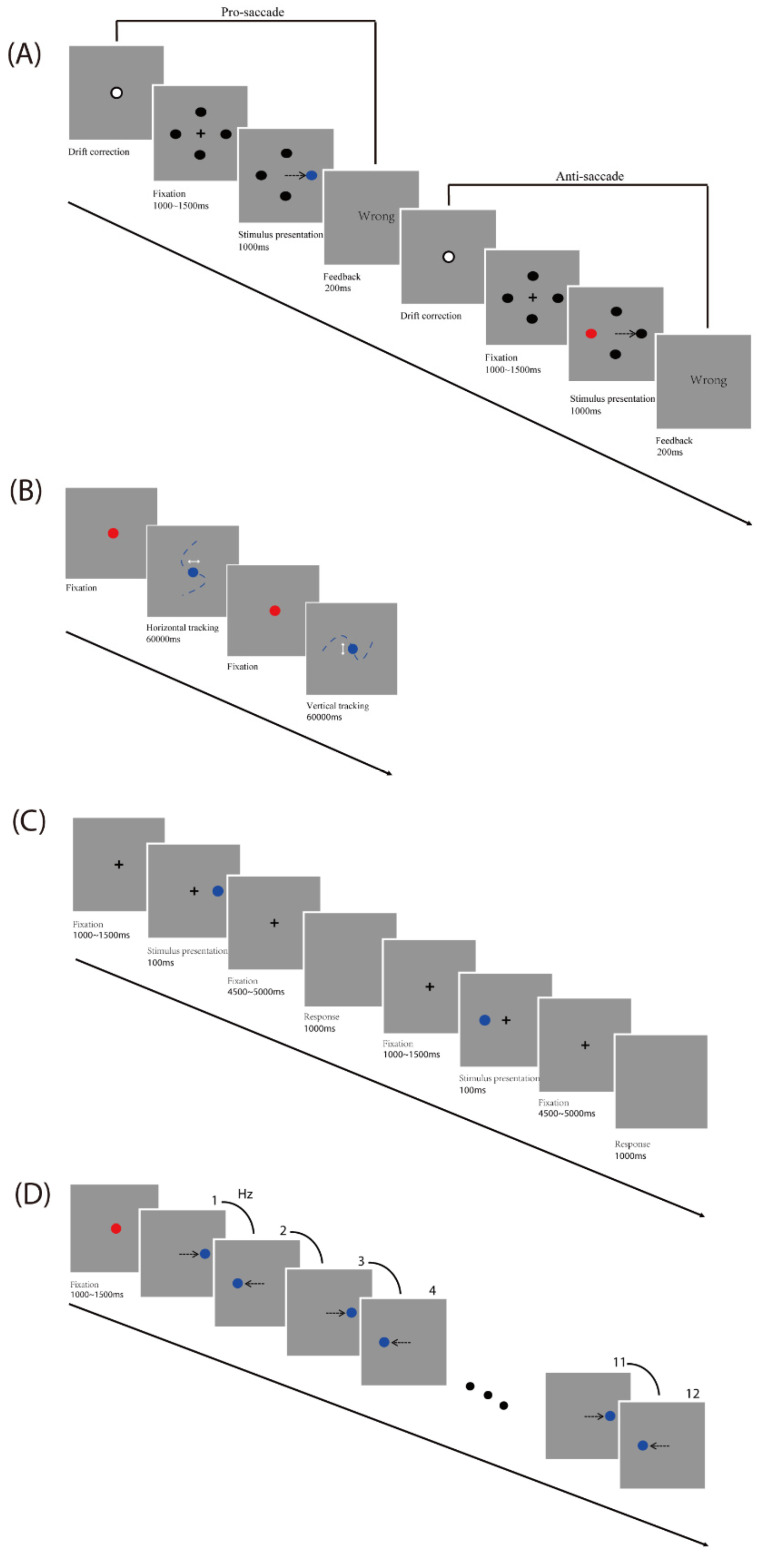
(**A**) pro- and anti-saccades task; (**B**) smooth pursuit task; (**C**) memory-guided task; (**D**) predictive saccade task. Red dot: target point of the anti-saccade (**A**), fixation point (**B**,**D**); blue dot: target point.

**Figure 2 biomedicines-13-00738-f002:**
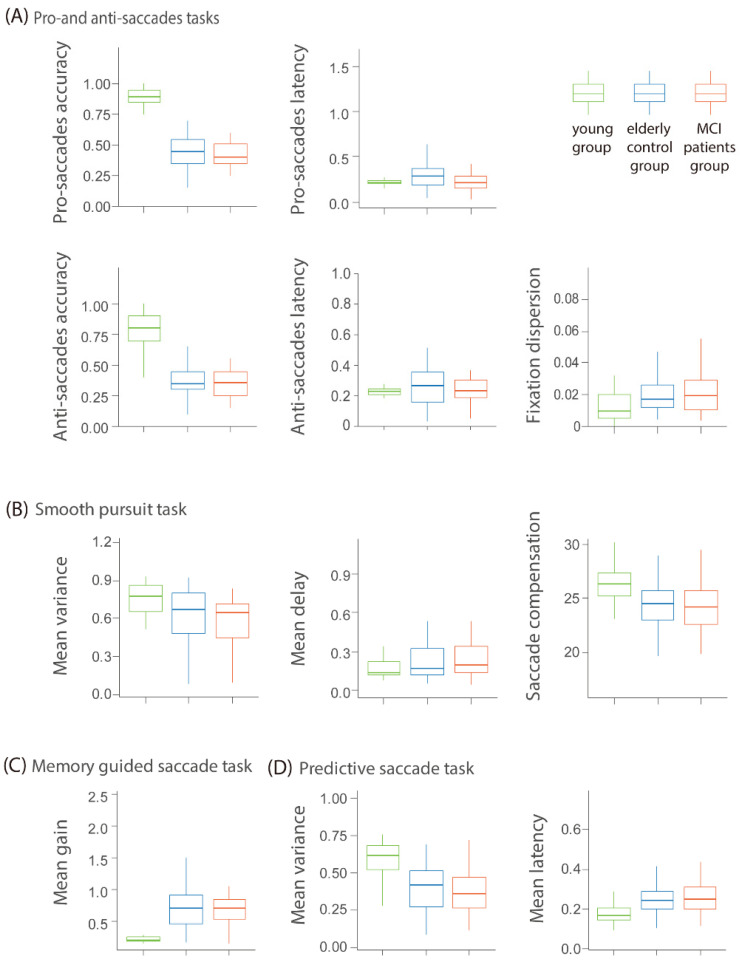
Difference among groups in eye movement tasks. (**A**) fixation dispersion; pro- and anti-saccades tasks’ accuracy (%) and latency (s); (**B**) smooth pursuit task: mean variance, mean delay(s), and saccade compensation; (**C**) memory-guided task: mean gain; (**D**) predictive saccade task: mean variance and latency(s).

**Figure 3 biomedicines-13-00738-f003:**
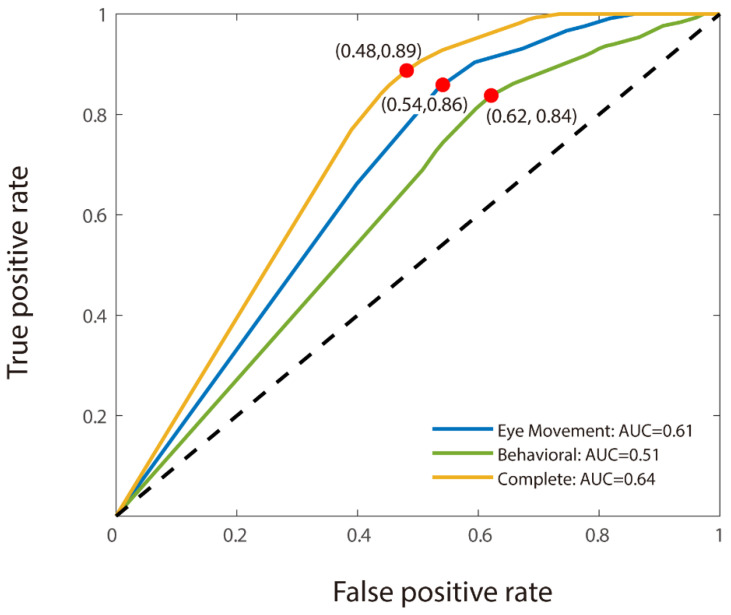
MCI patient detection power of models. Models based on eye movement features only (blue curve), behavioral features only (green curve), and combined features (included both eye movement and behavioral features) (orange curve) (AUC: 0.61, 0.51, and 0.64, respectively). The binary model based on combined features performs the best. AUC: area under receiver operating characteristic (ROC) curve.

**Table 1 biomedicines-13-00738-t001:** Data Input to CNN.

The Pro- and Anti-Saccades Tasks (10 Features)	Smooth Pursuit Task (12 Features)	Memory-Guided Task (5 Features)	Predictive Saccade Task (4 Features)	Behavioral Tasks (8 Features)
fixation dispersion	fixation dispersion	fixation dispersion	fixation dispersion	forward visuospatial task
mean accuracy	saccade compensation	mean gain	mean variance	backward visuospatial task
pro-saccades accuracy	mean variance	gain at 5°	mean latency	forward digit span task
anti-saccades accuracy	variance vertical at 0.25 Hz	gain at 10°	mean gain	backward digit span task
mean latency	variance vertical at 0.4 Hz	gain at 15°		Stroop consistent accuracy
pro-saccades latency	variance horizontal at 0.25 Hz			Stroop inconsistent accuracy
anti-saccades latency	variance horizontal at 0.4 Hz			Stroop consistent response time
mean gain	mean delay			Stroop inconsistent response time
pro-saccades gain	delay vertical at 0.25 Hz			
anti-saccades gain	delay vertical at 0.4 Hz			
	delay horizontal at 0.25 Hz			
	delay horizontal at 0.4 Hz			

**Table 2 biomedicines-13-00738-t002:** MCI Triple Classification Results.

	Accuracy (%)	Hit Rate (%)		
		Control	MCI	Young
Combined features	74.62 (3.71)	73.08 (12.63)	60.77 (12.51)	100.00 (0.00)
Eye movement features only	69.62 (5.27)	66.60 (7.41)	51.63 (15.29)	98.75 (3.95)
Behavioral features only	72.31 (7.43)	67.84 (12.34)	57.50 (11.62)	96.39 (5.83)
Combined features without pro- and anti-saccades task	68.85 (7.57)	70.07 (11.82)	51.50 (20.56)	94.90 (8.32)
Combined features without smooth pursuit task	70.39 (5.46)	61.73 (11.18)	58.60 (8.57)	100.00 (0.00)
Combined features without memory-guided saccade task	73.08 (8.70)	69.41 (15.64)	58.81 (19.83)	100.00 (0.00)
Combined features without prediction saccade task	78.48 (6.86)	73.70 (15.09)	62.64 (16.94)	100.00 (0.00)

Note: Data are expressed as Mean (Standard Deviation).

**Table 3 biomedicines-13-00738-t003:** MCI Binary Classification Results.

	Accuracy (%)	Hit Rate (%)	
		Control	MCI
Combined features	66.50 (8.51)	70.02 (11.15)	59.76 (13.90)
Eye movement features only	61.00 (6.15)	70.63 (10.56)	49.31 (13.15)
Behavioral features only	58.50 (4.12)	62.22 (13.77)	52.79 (15.88)
Combined features without pro- and anti-saccades task	61.00 (10.22)	62.08 (18.79)	56.97 (15.77)
Combined features without smooth pursuit task	62.50 (7.55)	68.21 (13.71)	54.58 (14.59)
Combined features without memory-guided saccade task	63.00 (6.75)	69.56 (11.03)	52.77 (21.40)
Combined features without prediction saccade task	66.00 (6.15)	70.78 (10.54)	59.88 (17.76)

Note: Data are expressed as Mean (Standard Deviation).

## Data Availability

The data supporting the conclusions of this article are available in an online repository (https://osf.io/sfcqj/ (accessed on 19 January 2025)).

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
