# Peer review of "Enhancing Mild Cognitive Impairment Auxiliary Identification Through Multimodal Cognitive Assessment with Eye Tracking and Convolutional Neural Network Analysis"

_biomedicines, 2025, doi:10.3390/biomedicines13030738_

Round 1

Reviewer 1 Report

Comments and Suggestions for Authors

Dear Authors.

The manuscript shows interesting findings that, with the implementation of advanced technology, contributes to obtaining early diagnostic indicators of mild cognitive impairment.
The introduction is well stated and leads to a necessary research problem of investigating how to specify preventive diagnoses in cognitive impairment, especially when they transition to Alzheimer's type dementia. It should be noted that it is not extensible to other dementias.
The methodology describes each procedure in detail with its corresponding justification according to the completed problem. Since the participants exceed N = 50, whether the data are distributed according to the normal curve with the Kolmogorov-Smirnov statistic could be examined to make statistical decisions such as Pearson's r (Line 310 - 312).
Line 108: It is suggested that the test they used to evaluate activities of daily living be mentioned. Is it the Barthel Test?
Line 87: The age ranges are wide, perhaps an analysis by segment could establish if age or another parameter could influence.
Line 180: There is a commendable evaluation process regarding cognitive tests. They present tasks of complexity that exceed those from traditional paper and pencil or self-report methodologies, enriching the record for analysis of the participants' cognition. On the other hand, processing a high accumulation of electronic records us a reaction time, execution time, and successes, among others, and also enriches the results for the respective analysis of each person's performance.
Line 221: Matlab, R, or Python offers the possibility of analysis that allows the visualization of trends that refer to the cognition of the participants, pertinent due to the high flow of data that is obtained from the electronic devices used.

The results show the tables and graphs reflect what was described regarding the findings despite their complexity. However, it suggests:
Line 310-312: Justify the normality in the distribution.
Line 350: The statement “All models were able to effectively distinguish young participants and achieved the lowest success rate of 94.90%” is difficult to understand in the line of the previous and subsequent rewording of this paragraph. If the success rate is lower, why does it indicate a percentage of 94.90%?
Line 380: In the ROC curve graphs, it would be interesting and suggested to examine the sensitivity and specificity calculations with a cut-off point can be established that distinguishes the three indicated models. This is also mentioned in the discussion in lines 418-420.

In discussion it is suggested:
Line 409 – 420: The history of visual attention is more related to dementia than to mild cognitive impairment. This affirmation may be suggested to relate this statement to the subject of DCL, despite what is mentioned below this paragraph. In this regard, it is suggested to review: DOI: 10.1186/s12877-023-04546-1
or DOI: 10.2174/1567205014666161213120807

Line 428 -431: It is suggested that this statement could be supported with some publication, given its importance in the context of cognitive assessment.
Line 434-446: What is stated in this paragraph is suggested to be supported. In Pubmed, you can find two articles that could be related to this evidence. Wilcockson et al., 2019; Opwonya et al., 2023.
Line 447-458: It also suggests supporting these assertions. On the other hand, saccadic movements could be related to working memory by mechanisms still not known. However, DCL is mostly linked to amnestic episodic memory or frontal memory and the evidence for this relationship with saccadic movements is scarce. Rather, the movements detected with eye tracking for this study are due to cognitive inhibition, which is key in executive function. There is evidence in this regard. A reasoned paragraph about it is suggested that could be an end to these interesting findings presented.

Author Response

Thank you so much for your time and effort in reviewing our manuscript. As requested, we have provided detailed responses to all the comments and questions. These responses can be found below. In addition, we have made the corresponding revisions and corrections to the manuscript.

Comments 1: The manuscript shows interesting findings that, with the implementation of advanced technology, contributes to obtaining early diagnostic indicators of mild cognitive impairment. The introduction is well stated and leads to a necessary research problem of investigating how to specify preventive diagnoses in cognitive impairment, especially when they transition to Alzheimer's type dementia. It should be noted that it is not extensible to other dementias. The methodology describes each procedure in detail with its corresponding justification according to the completed problem.

Since the participants exceed N = 50, whether the data are distributed according to the normal curve with the Kolmogorov-Smirnov statistic could be examined to make statistical decisions such as Pearson's r (Line 310 - 312).

Response 1: Thank you for your suggestion. we have conducted the Kolmogorov-Smirnov normality analysis as you suggested, and the variables were found to be non-normally distributed. Given this, we used Spearman's correlation for the analysis. For the specific results, please refer to Table S3. The content and results are detailed in the response below.

Comments 2: Line 108: It is suggested that the test they used to evaluate activities of daily living be mentioned. Is it the Barthel Test?

Response 2: We appreciate your attention to the details of our study. To clarify, the test we used to evaluate activities of daily living (ADL) is indeed the Barthel Test. We have employed the Barthel Index to assess the functional status and independence of participants in their daily activities. This widely recognized tool provides a comprehensive evaluation of ADL, making it suitable for our research objectives. This information has been clearly mentioned in the manuscript(Page 3 Line 117).  We hope it helps readers better understand the methods used in our research.

Comments 3: Line 87: The age ranges are wide, perhaps an analysis by segment could establish if age or another parameter could influence.

Response 3: Considering that the average and median age of MCI patients is 70 years, the patients are divided into two groups: those aged 70 or younger and those older than 70. The data from these two groups are then input into the model for binary classification. The results show an average accuracy of 61.25%, with a classification accuracy of 52.83% for patients aged 70 or younger and 69.50% for patients older than 70. This suggests that age may influence the cognitive status of patients, leading to differences in cognitive abilities across different age groups.

To further verify this, we conduct two triple classification experiments. The first triple classification includes: MCI patients aged 70 or younger, elderly control group, and young individuals; the second triple classification includes: MCI patients older than 70, elderly control group, and young individuals. The results of the first triple classification are as follows: the average accuracy is 81.90%, the accuracy for the elderly control group is 89.97%, the accuracy for MCI patients aged 70 or younger is 39.50%, and the accuracy for young individuals is 100%. The results of the second triple classification are as follows: the average accuracy is 77.83%, the accuracy for the elderly control group is 83.61%, the accuracy for MCI patients older than 70 is 34.37%, and the accuracy for young individuals is 100%.

Although the two groups of MCI patients may have differences in cognitive functions due to age, these potential differences were not detected in the subsequent triple classification model tests. Neither MCI patients aged 70 or younger nor those older than 70 show a clear trend that can distinguish them from the elderly control group. This does not mean that MCI patients cannot be distinguished from the elderly control group, since when we combine the two groups of MCI patients, the classification accuracy for MCI patients exceeds 60%, as shown in the main body of our paper. The above results are largely due to the reduced sample size caused by splitting the MCI dataset. The decrease in the amount of MCI data makes it difficult for the CNN model to learn effective features, which hinders its ability to effectively distinguish normal elderly individuals from MCI patients.

Comments 4: Line 180: There is a commendable evaluation process regarding cognitive tests. They present tasks of complexity that exceed those from traditional paper and pencil or self-report methodologies, enriching the record for analysis of the participants' cognition. On the other hand, processing a high accumulation of electronic records us a reaction time, execution time, and successes, among others, and also enriches the results for the respective analysis of each person's performance.

Response 4: We thank the appreciation of the reviewer indeed.

Comments 5: Line 221: Matlab, R, or Python offers the possibility of analysis that allows the visualization of trends that refer to the cognition of the participants, pertinent due to the high flow of data that is obtained from the electronic devices used.

Response 5: We appreciate your feedback on the description of using Matlab, R, or Python for data analysis. We agree that these tools are essential for visualizing trends related to participants' cognition, especially given the high volume of data obtained from electronic devices.

Comments 6: The results show the tables and graphs reflect what was described regarding the findings despite their complexity. However, it suggests:

Line 310-312: Justify the normality in the distribution.

Response 6: As previously stated, the data were found to be non-normally distributed. Therefore, we opted for Spearman's correlation analysis. Although there were changes in the results, the overall relationship between eye movement performance and behavioral test performance remained consistent. For the specific results, please refer to Table S3. Meanwhile, the content on Page 10 Line 357-358,362 of the manuscript has also been adjusted accordingly.

Comments 7: Line 350: The statement “All models were able to effectively distinguish young participants and achieved the lowest success rate of 94.90%” is difficult to understand in the line of the previous and subsequent rewording of this paragraph. If the success rate is lower, why does it indicate a percentage of 94.90%?

Response 7: Firstly, the 94.90% figure mentioned specifically refers to the hit rate for identifying young participants. Given that young participants generally exhibit significantly higher cognitive abilities compared to the elderly groups (which include both the control and MCI groups), achieving a high hit rate for this group aligns with our expectations.

The term "accuracy," as noted in Table 2, Column 2, represents the overall accuracy of the model. It is calculated using the formula: Accuracy = Number of Correct Predictions / Total Number of Predictions, "Number of Correct Predictions" includes all participants correctly classified into young, MCI, or control groups, and "Total Number of Predictions" refers to the total number of participants. Thus, accuracy reflects the model's ability to correctly classify all three groups collectively. On the other hand, the figures like 94.90% in the subsequent columns of Table 2 and Table 3 represent the hit rates for each group individually. These hit rates indicate the model's ability to correctly identify each specific group from the overall pool of participants.

In summary, while "accuracy" captures the model's overall classification performance across all groups, "hit rate" measures the model's capability to distinguish a particular group from the rest. The cognitive distinctions between MCI and control groups are more nuanced compared to those of the young group, which naturally leads to lower hit rates for these groups. The accuracy figure, being an average across all groups, is consequently lower than the hit rate for the young group.

Comments 8: Line 380: In the ROC curve graphs, it would be interesting and suggested to examine the sensitivity and specificity calculations with a cut-off point can be established that distinguishes the three indicated models. This is also mentioned in the discussion in lines 418-420.

Response 8: We have carefully considered your feedback and have updated Figure 3 to include cut-off points for the three indicated models. These cut-off points were determined based on the optimal balance between sensitivity and specificity for each model, as suggested. In the revised Figure 3 (Page 12), the cut-off points are clearly marked on the ROC curves for each model, providing a visual representation of the threshold values that best distinguish between the groups. The corresponding sensitivity and specificity values for these cut-off points are also detailed in the figure legend.

Comments 9: In discussion it is suggested:

Line 409 – 420: The history of visual attention is more related to dementia than to mild cognitive impairment. This affirmation may be suggested to relate this statement to the subject of DCL, despite what is mentioned below this paragraph. In this regard, it is suggested to review:

DOI: 10.1186/s12877-023-04546-1 or DOI: 10.2174/1567205014666161213120807

Response 9: Thank you for your insightful comments. We appreciate your suggestion to relate our findings to the subject of DCL. However, we must clarify that the term “DCL” was not explicitly defined in the context of our study, and we are uncertain about its specific meaning in this context. Based on our understanding, we speculate that DCL might refer to “Direct Cognitive Level,” which could be related to the severity or type of cognitive impairment.

We agree that visual attention models have potentials in predicting dementia, particularly in the progression from MCI to AD. Our study builds on this foundation by demonstrating a strong link between eye movement abnormalities and specific cognitive functions. For instance, dual-task eye-tracking and gait analysis have achieved high accuracy in distinguishing MCI from dementia. However, identifying MCI in the general population remains challenging due to the heterogeneity of cognitive impairments. In our study, we focused on the role of cognitive inhibition and executive function, which are key components in tasks requiring high cognitive load. These functions are closely related to the eye movement patterns we observed, particularly in tasks involving memory and attention.

Our eye-tracking data primarily captured features related to cognitive inhibition, which is essential for maintaining focus and filtering out irrelevant information. This is particularly relevant in tasks that require high cognitive load, such as those involving working memory and executive function. While the relationship between saccadic movements and working memory is still not fully understood, our findings suggest that cognitive inhibition plays a significant role in the observed eye movement patterns.

In addition to the findings presented, our study suggests that the eye-tracking data captured in this research are primarily reflective of cognitive inhibition, a key component of executive function under high cognitive load. While saccadic movements have been hypothesized to relate to working memory through mechanisms that are still not fully understood, our results indicate that these movements are more closely associated with cognitive inhibition. This finding is consistent with studies showing the importance of cognitive inhibition in maintaining focus and filtering out irrelevant information, particularly in tasks requiring high cognitive load. Future research should explore the potential links between saccadic movements, cognitive inhibition, and DCL to further elucidate these mechanisms. (Page 13 Line 483-492)

Comments 10: Line 428 -431: It is suggested that this statement could be supported with some publication, given its importance in the context of cognitive assessment.

Response 10: Thank you for your suggestion regarding the need to support our statement with relevant publications. We have reviewed the literature and included specific references (see Page 13 Line 503) to substantiate the assertion that motor limitations and diverse educational backgrounds can impact cognitive assessments in older adults.

Comments 11: Line 434-446: What is stated in this paragraph is suggested to be supported. In Pubmed, you can find two articles that could be related to this evidence. Wilcockson et al., 2019; Opwonya et al., 2023.

Response 11: We do thank the reviewer for listing these articles, all of which are useful and have helped us to refine our literature review. We have added these articles as references and revised multiple places of the manuscript (see Page 14 Line 508).

Comments 12: Line 447-458: It also suggests supporting these assertions. On the other hand, saccadic movements could be related to working memory by mechanisms still not known. However, DCL is mostly linked to amnestic episodic memory or frontal memory and the evidence for this relationship with saccadic movements is scarce. Rather, the movements detected with eye tracking for this study are due to cognitive inhibition, which is key in executive function. There is evidence in this regard. A reasoned paragraph about it is suggested that could be an end to these interesting findings presented.

Response 12: Thank you for your insightful comments and suggestions regarding the relationship between saccadic movements, cognitive functions, and the broader context of DCL, which we assume stands for " Direct Cognitive Level " based on the context. We appreciate your guidance on how to better contextualize our findings.

We agree that eye movement abnormalities, particularly in memory-related tasks, can serve as valuable biomarkers for differentiating between MCI subtypes. Specifically, amnestic MCI (aMCI) patients exhibit distinct eye movement patterns compared to non-amnestic MCI (naMCI) patients, highlighting the importance of considering MCI subtypes in diagnostic accuracy (Ma et al., 2025).

We agree that our findings support the assertion that multi-dimensional parameters, including eye movement and behavioral measures, are valuable for early detection of MCI. The inclusion of various eye movement tasks and behavioral features in our model aims to capture the complexity of cognitive impairments and improve diagnostic sensitivity.

You correctly pointed out that the relationship between saccadic movements and working memory is still not fully understood. While some studies suggest potential links, the mechanisms remain unclear. In our study, we focused on the role of cognitive inhibition, which is a key component of executive function and is closely related to DCL. Cognitive inhibition is essential for maintaining focus and filtering out irrelevant information, which is particularly relevant in tasks requiring high cognitive load.

We acknowledge that DCL is primarily associated with amnestic episodic memory and frontal memory functions. However, the evidence linking these memory functions directly to saccadic movements is limited. Instead, our eye-tracking data primarily captured movements related to cognitive inhibition, which is a critical aspect of executive function under high cognitive load. This finding aligns with studies showing that cognitive inhibition plays a significant role in early cognitive decline.

Reviewer 2 Report

Comments and Suggestions for Authors

The reviewed work is devoted to the development of methods for diagnosing Mild Cognitive Impairment (MCI), which is undoubtedly important due to the fact that MCI is often the beginning of Alzheimer's disease. And although the currently used diagnostic methods cope well with diagnosing MCI, the development of modern approaches is necessary for early diagnosis. The authors presented, in my opinion, a pilot project, which can be further improved with the achievement of more accurate indicators in diagnosing MCI.

Some questions and comments arose during the review of the manuscript.

- In the Introduction, the authors mention some limitations of the MMSE and MoCA. It is necessary to provide known quantitative estimates for these systems.

- Initially, 147 people participated in the study, later their number was reduced to 128. 19 people were unable to complete the tests: 12 people from the control group (17.3%) and 9 from the MCI group (14.9%). Is this due to the complexity of the tests? How do the authors explain such a large percentage?

- The study involves elderly patients, therefore the time of testing and their duration may affect the results. It is necessary to describe the conditions of testing in more detail.

- The studies were conducted on mixed groups of individuals. Why was this study design chosen, and not separately among men and women?

- The patients' ages ranged from 56 to 87 years, which is a large age range during which mental abilities decline significantly with normal aging. Perhaps the authors could have increased the sensitivity of the method by decreasing the age ranges (60-70 and 70-80, etc.).

- It is unclear whether the level of education and intelligence of the patients before the decline in cognitive functions was taken into account and whether the control and experimental groups were the same in these parameters.

- Files Table S, Table S2, Table S3 are missing, which complicates the analysis of the results.

- It is necessary to check the developed method, when, based on a certain group of patients with a presumptive diagnosis of MCI, using this method, MCI is diagnosed (or not), which is then verified by standard methods.

Comments on the Quality of English Language

The English could be improved 

Author Response

Many thanks for reviewing our manuscript. Please find the detailed responses below and the corresponding revisions in the re-submitted files.

Comments 1: The reviewed work is devoted to the development of methods for diagnosing Mild Cognitive Impairment (MCI), which is undoubtedly important due to the fact that MCI is often the beginning of Alzheimer's disease. And although the currently used diagnostic methods cope well with diagnosing MCI, the development of modern approaches is necessary for early diagnosis. The authors presented, in my opinion, a pilot project, which can be further improved with the achievement of more accurate indicators in diagnosing MCI.

Some questions and comments arose during the review of the manuscript.

Response 1: Thank you for your valuable feedback. We are grateful for your recognition of the importance of developing methods for diagnosing MCI, especially given its role as a potential precursor to Alzheimer's disease. We fully agree that while current diagnostic methods are effective to some extent, the development of modern approaches is essential for earlier and more accurate diagnosis.

Comments 2: In the Introduction, the authors mention some limitations of the MMSE and MoCA. It is necessary to provide known quantitative estimates for these systems.

Response 2: We appreciate the reviewer's feedback. In response, we have revised the Introduction to highlight the key limitations of traditional cognitive assessment tools like the MMSE and MoCA. For example, we now mention the impact of educational bias on these tools, the issue of scoring subjectivity, and the ceiling effects of the MMSE. These revisions emphasize the need for more objective and education-adjusted assessment methods. For detailed information, please refer to Page 2 Line 48-50,53-59.

Comments 3: Initially, 147 people participated in the study, later their number was reduced to 128. 19 people were unable to complete the tests: 12 people from the control group (17.3%) and 9 from the MCI group (14.9%). Is this due to the complexity of the tests? How do the authors explain such a large percentage?

Response 3: We would like to explain that our study was conducted as part of a broader elderly care program organized by the government. In addition to the cognitive assessments related to our study, participants were also required to undergo evaluations for anxiety, depression, and physical health. The overall process was time-consuming, which contributed to the higher attrition rate. Moreover, we allowed participants the option to withdraw from the study at any time, respecting their autonomy. While these individuals completed most of the tasks, they were unable to finish all the required assessments. As a result, their incomplete data were excluded from the final analysis. The incomplete data in both groups occurred and should not be attributed to the complexity of our study tasks. Instead, it is more likely related to the broader context of data collection, including participants' personal preferences.

Comments 4: The study involves elderly patients, therefore the time of testing and their duration may affect the results. It is necessary to describe the conditions of testing in more detail.

Response 4: Thank you for your feedback. We understand the potential impact of testing duration and fatigue on the results in elderly patients. We have implemented several measures to address this issue. Specifically, we incorporated rest periods during tasks such as the smooth pursuit task, arranged rest periods between different blocks of the task, included longer rest periods in both behavioral and eye movement tests, and individualized the testing time for each participant. These measures were designed to minimize the impact of fatigue and ensure the comfort of the participants. For more detailed information, please refer to the revised Procedure section of the manuscript (Page 6 Line230-241).

Comments 5: The studies were conducted on mixed groups of individuals. Why was this study design chosen, and not separately among men and women?

Response 5: First, the main goal of this study is to explore the role of eye-tracking and behavioral features in the auxiliary identification of MCI, rather than specifically investigating the impact of gender differences on MCI. Therefore, the design of a mixed-gender group aligns with our research objectives, helping to grasp the overall characteristics of MCI patients and enhancing the representativeness and generalizability of the results.

Secondly, considering the sample size limitation, we are concerned with that, if male and female MCI patients are studied separately, the sample size for each gender group might be too small compared to the elderly control group. This could result in the deep learning model failing to learn the key features of MCI patients and, thus misclassifying them as part of the elderly control group.

Comments 6: The patients' ages ranged from 56 to 87 years, which is a large age range during which mental abilities decline significantly with normal aging. Perhaps the authors could have increased the sensitivity of the method by decreasing the age ranges (60-70 and 70-80, etc.).

Response 6: Considering that the average and median age of MCI patients is 70 years, the patients are divided into two groups: those aged 70 or younger and those older than 70. The data from these two groups are then input into the model for binary classification. The results show an average accuracy of 61.25%, with a classification accuracy of 52.83% for patients aged 70 or younger and 69.50% for patients older than 70. This suggests that age may influence the cognitive status of patients, leading to differences in cognitive abilities across different age groups.

To further verify this, we conduct two triple classification experiments. The first triple classification includes: MCI patients aged 70 or younger, elderly control group, and young individuals; the second triple classification includes: MCI patients older than 70, elderly control group, and young individuals. The results of the first triple classification are as follows: the average accuracy is 81.90%, the accuracy for the elderly control group is 89.97%, the accuracy for MCI patients aged 70 or younger is 39.50%, and the accuracy for young individuals is 100%. The results of the second triple classification are as follows: the average accuracy is 77.83%, the accuracy for the elderly control group is 83.61%, the accuracy for MCI patients older than 70 is 34.37%, and the accuracy for young individuals is 100%.

Although the two groups of MCI patients may have differences in cognitive functions due to age, these potential differences were not detected in the subsequent triple classification model tests. Neither MCI patients aged 70 or younger nor those older than 70 show a clear trend that can distinguish them from the elderly control group. This does not mean that MCI patients cannot be distinguished from the elderly control group, since when we combine the two groups of MCI patients, the classification accuracy for MCI patients exceeds 60%, as shown in the main body of our paper. The above results are largely due to the reduced sample size caused by splitting the MCI dataset. The decrease in the amount of MCI data makes it difficult for the CNN model to learn effective features, which hinders its ability to effectively distinguish normal elderly individuals from MCI patients.

Comments 7: It is unclear whether the level of education and intelligence of the patients before the decline in cognitive functions was taken into account and whether the control and experimental groups were the same in these parameters.

Response 7: Thank you for raising this important issue regarding the consideration of education and intelligence levels in our study. For the young adult group, all participants were undergraduate students recruited from a university setting, ensuring a relatively homogeneous educational background (i.e., university-level education). This aspect was not a concern for this specific cohort. However, for the older adult group, we acknowledge that we did not formally assess their education and intelligence levels prior to cognitive decline. We categorized their educational background based on self-reports, which is a limitation given the diverse and often non-standardized educational experiences of this age group. Due to China's historical background, many older adults received non-formal and non-systematic education, making it challenging to quantify their education in terms of years.

While we recognize that education could potentially influence cognitive performance, our primary focus was on evaluating whether eye-tracking technology could effectively match clinical MCI diagnoses made by physicians. From this perspective, the specific impact of education was not our main concern. Nevertheless, we agree that not exploring this aspect in detail is a limitation. Future work could further investigate these factors to provide a more comprehensive understanding.

Comments 8: Files Table S, Table S2, Table S3 are missing, which complicates the analysis of the results.

Response 8: Thank you for bringing this to our attention. We apologize for having forgotten to submit the supplementary tables (Table S, Table S2, Table S3) and the inconvenience caused by the missing during the submission process. These tables were initially uploaded as supplementary files, but we understand that their absence complicates the analysis of our results.

To address this issue, we have incorporated the relevant tables directly into the manuscript in the next revision (see the Supplementary Analyses at Page 19). This shall ensure that all necessary information is readily accessible to reviewers and readers, facilitating a more seamless evaluation of our study. We appreciate your patience and understanding, and we will ensure that the manuscript is complete and self-contained in our next submission.

Comments 9: It is necessary to check the developed method, when, based on a certain group of patients with a presumptive diagnosis of MCI, using this method, MCI is diagnosed (or not), which is then verified by standard methods.

Response 9: We fully agree with you regarding the necessity of validating our developed method for diagnosing MCI. This validation process is crucial to ensure the reliability and accuracy of our approach when compared to standard diagnostic methods. While our current study lays the foundation for developing this new method, we recognize that its validation using a cohort of patients with a presumptive diagnosis of MCI is an essential next step.

However, we must acknowledge the significant challenges associated with recruiting and assessing MCI patients for such validation studies. MCI is characterized by subtle cognitive deficits that often go unnoticed in the early stages. This makes it difficult to identify suitable participants who are in the early stages of the condition. Moreover, the need for comprehensive assessments, including cognitive tests, neuroimaging, and biomarker analyses, adds to the complexity and resource demands of such studies.

Despite these challenges, we plan to conduct a comprehensive validation study where our method will be applied to a group of patients who have been preliminarily diagnosed with MCI. By using our method to diagnose MCI in these patients and then verifying the results with MoCA, we hope to demonstrate the effectiveness of our approach. Additionally, we will explore the integration of other diagnostic modalities, including neuroimaging techniques like MRI or PET scans, to further corroborate our findings. This multi-faceted approach will allow us to comprehensively evaluate the performance of our method and confirm its robustness in clinical settings.

We believe that this validation process will not only enhance the credibility of our method but also contribute valuable insights to the field of MCI diagnosis. By comparing our results with these standard methods, we aim to demonstrate the potential of our approach to provide a more objective and accurate assessment of cognitive impairment. We look forward to incorporating these validation steps in our future research and sharing the outcomes with the broader scientific community.

Reviewer 3 Report

Comments and Suggestions for Authors

This study investigates a novel approach for identifying Mild Cognitive Impairment (MCI) by integrating eye-tracking technology and convolutional neural networks (CNNs). Traditional assessments like the Mini-Mental State Examination (MMSE) and the Montreal Cognitive Assessment (MoCA) have limitations, necessitating more objective and scalable methods. The study involved 128 participants (40 MCI patients, 57 elderly controls, and 31 young adults) who completed four eye movement tasks (pro- and anti-saccades, smooth pursuit, memory-guided saccades, and predictive saccades) and two cognitive tests assessing inhibitory control and working memory. A CNN-based classification model was developed using 31 eye movement features and 8 behavioural features. Results showed that a multimodal model combining eye-tracking and behavioural data achieved higher accuracy (74.62%) compared to models using only one feature type, with eye-tracking data alone performing better in distinguishing younger from older participants but being less effective in differentiating MCI from healthy elderly individuals. The findings highlight the potential of integrating multimodal data for early MCI detection, suggesting that eye-tracking combined with machine learning could serve as a valuable tool for community-based cognitive screening.

Appraisal and comments

This study presents an interesting and innovative approach to identifying Mild Cognitive Impairment (MCI) by integrating eye-tracking technology and convolutional neural networks (CNNs). By combining eye movement features with behavioural data, the authors demonstrate the potential of a multimodal model for early detection, achieving higher classification accuracy than single-feature models.

 Given the moderate accuracy in distinguishing MCI from healthy elderly individuals and the complexity of cognitive decline, future studies could benefit from incorporating microsaccade analysis. Microsaccades, as fine involuntary eye movements linked to attention and cognitive processing (Rolfs, 2009). Research suggests microsaccades are sensitive indicators of cognitive load in pro-anti saccade tasks (Dalmaso et al., 2020) and neurodegenerative changes (Kapoula et al., 2014), making them a promising addition to existing eye-tracking metrics. Including microsaccade metrics alongside saccadic and pursuit-based features could enhance model sensitivity and contribute to a more comprehensive cognitive assessment framework.

I, therefore, suggest expanding the general discussion to report this possibility.

References

Dalmaso, M., Castelli, L., & Galfano, G. (2020). Microsaccadic rate and pupil size dynamics in pro-/anti-saccade preparation: the impact of intermixed vs. blocked trial administration. Psychological Research, 84(5), 1320-1332.

Kapoula, Z., Yang, Q., Otero-Millan, J., Xiao, S., Macknik, S. L., Lang, A., ... & Martinez-Conde, S. (2014). Distinctive features of microsaccades in Alzheimer’s disease and in mild cognitive impairment. Age36, 535-543.

Rolfs, M. (2009). Microsaccades: small steps on a long way. Vision research, 49(20), 2415-2441.

Author Response

Thanks so much for reviewing our manuscript. Here are our responses to your comments. Please find the detailed responses below and the corresponding revisions in the re-submitted files.

Comments 1: This study investigates a novel approach for identifying Mild Cognitive Impairment (MCI) by integrating eye-tracking technology and convolutional neural networks (CNNs). Traditional assessments like the Mini-Mental State Examination (MMSE) and the Montreal Cognitive Assessment (MoCA) have limitations, necessitating more objective and scalable methods. The study involved 128 participants (40 MCI patients, 57 elderly controls, and 31 young adults) who completed four eye movement tasks (pro- and anti-saccades, smooth pursuit, memory-guided saccades, and predictive saccades) and two cognitive tests assessing inhibitory control and working memory. A CNN-based classification model was developed using 31 eye movement features and 8 behavioural features. Results showed that a multimodal model combining eye-tracking and behavioural data achieved higher accuracy (74.62%) compared to models using only one feature type, with eye-tracking data alone performing better in distinguishing younger from older participants but being less effective in differentiating MCI from healthy elderly individuals. The findings highlight the potential of integrating multimodal data for early MCI detection, suggesting that eye-tracking combined with machine learning could serve as a valuable tool for community-based cognitive screening.

Response 1: Thank you for your appreciation and insightful comments on our study. We appreciate your recognition of the innovative approach we used to identify MCI by integrating eye-tracking and CNNs. We are encouraged by the results, which highlight the value of integrating multiple data sources to improve diagnostic accuracy. We plan to expand our dataset and refine our model in future work to further validate our approach. Thank you again for your valuable feedback.

Comments 2:

Appraisal and comments

This study presents an interesting and innovative approach to identifying Mild Cognitive Impairment (MCI) by integrating eye-tracking technology and convolutional neural networks (CNNs). By combining eye movement features with behavioural data, the authors demonstrate the potential of a multimodal model for early detection, achieving higher classification accuracy than single-feature models.

Response 2: Thank you for your appreciation and insightful comments on our study. We are encouraged by your recognition of the innovative approach we have taken to identify MCI by integrating eye-tracking technology and CNNs. Your insights regarding the potential benefits of incorporating microsaccade analysis are particularly valuable and align well with our goals of enhancing early detection accuracy.

Comments 3: Given the moderate accuracy in distinguishing MCI from healthy elderly individuals and the complexity of cognitive decline, future studies could benefit from incorporating microsaccade analysis. Microsaccades, as fine involuntary eye movements linked to attention and cognitive processing (Rolfs, 2009). Research suggests microsaccades are sensitive indicators of cognitive load in pro-anti saccade tasks (Dalmaso et al., 2020) and neurodegenerative changes (Kapoula et al., 2014), making them a promising addition to existing eye-tracking metrics. Including microsaccade metrics alongside saccadic and pursuit-based features could enhance model sensitivity and contribute to a more comprehensive cognitive assessment framework.

I, therefore, suggest expanding the general discussion to report this possibility.

Response 3: We agree that given the moderate accuracy in distinguishing MCI from healthy elderly individuals, exploring additional metrics such as microsaccades could significantly improve our model's sensitivity and provide deeper insights into cognitive decline. Including microsaccade metrics alongside our existing saccadic and pursuit-based features could indeed enhance our model's ability to detect subtle cognitive impairments, contributing to a more comprehensive cognitive assessment framework.

In response to your suggestion, we have expanded General Discussion section to highlight the potential benefits of incorporating microsaccade analysis (Page 14 Line 540-549). We have also referenced the studies you mentioned to provide a solid theoretical foundation for this proposed enhancement. This addition will underscore the importance of exploring new metrics to improve early detection of MCI and align our work with the latest advancements in the field.

Reviewer 4 Report

Comments and Suggestions for Authors

The manuscript explores the use of a multimodal cognitive assessment approach to enhance Mild Cognitive Impairment (MCI) auxiliary identification. By integrating eye-tracking data and Convolutional Neural Network (CNN) analysis, the study aims to improve the diagnostic accuracy compared to traditional tools like the Mini-Mental State Examination (MMSE) and the Montreal Cognitive Assessment (MoCA). The study involved 128 participants and tested multiple feature sets to classify MCI patients with an accuracy of 74.62% when using combined eye-tracking and behavioral features. The results indicate that a multimodal approach is superior to single-feature models, suggesting potential applications for early MCI detection in community settings.

1. Please provide more information regarding the CNN setup. For example:

(1) The manuscript lacks a clear explanation of the CNN’s input format. Specifically, are eye movement features provided as videos or scalar features (e.g., pro-saccades accuracy, fixation dispersion, etc.) as seen in Figure 2? How are behavioral and eye-tracking features merged into CNN input?

(2) CNN is typically used to extract spatial features from structured input like images and time sequence data. However, if the input features are already scalar values, what specific advantage does convolution provide in this case?

 (3) Given that the model contains 32 hidden layers, the statement “The three fully connected layers have outputs of 4096, 256, and the number of classes” is unclear.

2. The acronyms MCI and CNN should be fully expanded in the title as they may have different meanings in other research domains. This ensures clarity for a broader audience.

3. The study only evaluates CNN, but a comparison with other machine learning models could provide valuable insight into the effectiveness of different architectures. (1) Given that CNNs are designed for extracting spatial dependencies, would a fully connected neural network (FCNN) without convolutions yield similar results if the input consists of scalar features rather than spatial data?

(2) Exploring other approaches, such as Support Vector Machines (SVM) and Random Forest, would strengthen the manuscript’s claims about the superiority of CNN in this context.

Author Response

We are grateful for your time spent reviewing this manuscript. The detailed responses are listed below, and the revised manuscript with the corresponding changes has been resubmitted.

Comments 1: The manuscript explores the use of a multimodal cognitive assessment approach to enhance Mild Cognitive Impairment (MCI) auxiliary identification. By integrating eye-tracking data and Convolutional Neural Network (CNN) analysis, the study aims to improve the diagnostic accuracy compared to traditional tools like the Mini-Mental State Examination (MMSE) and the Montreal Cognitive Assessment (MoCA). The study involved 128 participants and tested multiple feature sets to classify MCI patients with an accuracy of 74.62% when using combined eye-tracking and behavioral features. The results indicate that a multimodal approach is superior to single-feature models, suggesting potential applications for early MCI detection in community settings.

Response 1: We thank the reviewer for their comments. First, we have revised Section 2.4 in Methods and refined the description of the CNN model architecture, including the format of input features. Second, we modified the Results section and added Section 3.5, which compares CNN with other models. In this section, we compared the performance of CNN with that of SVM, Random Forest, and FCNN in distinguishing MCI patients. In Section 4, the Discussion, we elaborated on the important role of the convolutional operations in the CNN model for distinguishing MCI.

Please provide more information regarding the CNN setup. For example:

Comments 2: The manuscript lacks a clear explanation of the CNN’s input format. Specifically, are eye movement features provided as videos or scalar features (e.g., pro-saccades accuracy, fixation dispersion, etc.) as seen in Figure 2? How are behavioral and eye-tracking features merged into CNN input?

Response 2: Eye-tracking and behavioral features are not provided to the CNN in video format, but are extracted as scalar features from four eye-movement tasks (pro- and anti-saccades task, smooth pursuit task, memory-guided saccades task, and predictive saccades task) and two cognitive tasks (Stroop Task and Working Memory Tasks) using Matlab. These features include 31 eye movement features and 8 behavioral features as seen in Table 1. We concatenate the eye-tracking features and behavioral features into a single feature vector, meaning that each participant has 39 features that are fed into the model. For details, please see Page 6-7 Line 243-253.

Comments 3: CNN is typically used to extract spatial features from structured input like images and time sequence data. However, if the input features are already scalar values, what specific advantage does convolution provide in this case?

Response 3: We compare the classification performance of CNN and FCNN (Fully Connected Neural Network) in distinguishing MCI patients. The FCNN removes the feature extraction components of the CNN model (including the convolutional layers and pooling layers), retaining only the fully connected layers, activation functions, and Dropout layers. The number of output features for all fully connected layers is the same as in the CNN. The results show that the FCNN has lower average accuracy (binary classification: 57.50%; triple classification: 67.69%) and MCI accuracy (binary classification: 51.94%; triple classification: 47.09%) compared to CNN (which has average accuracies of 66.50% for binary classification and 74.62% for triple classification, and MCI accuracies of 59.76% for binary classification and 60.77% for triple classification).

We believe that this phenomenon may be attributed to the fact that the convolutional layers of CNN can extract high-dimensional features from the data, and these features, compared to individual eye-tracking features (such as fixation dispersion), are more indicative of the participants' cognitive states. CNN can learn the local patterns among these features, thereby extracting feature combinations that are more effective in distinguishing MCI patients. For details, please see the Discussion section.

Comments 4: Given that the model contains 32 hidden layers, the statement “The three fully connected layers have outputs of 4096, 256, and the number of classes” is unclear.

Response 4: We have provided more details in this part to make the description of the model clearer. The model we constructed consists of two one-dimensional convolutional layers and two one-dimensional pooling layers stacked alternately, followed by three fully connected layers for the output. The number of output channels for the first convolutional layer is 16, while the number of output channels for the second convolutional layer is 32. The first fully connected layer outputs 4096 features to help the network capture more complex data characteristics, while the second layer outputs 256 features to simplify the rich feature information from the previous layer and focuses on more representative features. The third fully connected layer is the final output layer, with the number of output features corresponding to the number of categories (2 for binary classification, 3 for triple classification), representing the final classification result. For details, see Page 7 Line 265-272.

Comments 5: The acronyms MCI and CNN should be fully expanded in the title as they may have different meanings in other research domains. This ensures clarity for a broader audience.

Response 5: Thank you for the suggestion. We have made the revision in the manuscript (Page 1 Line 2-4).

Comments 6: The study only evaluates CNN, but a comparison with other machine learning models could provide valuable insight into the effectiveness of different architectures. (1) Given that CNNs are designed for extracting spatial dependencies, would a fully connected neural network (FCNN) without convolutions yield similar results if the input consists of scalar features rather than spatial data?

Response 6: We have added the comparison between the classification performances of CNN and FCNN in order to explore the importance of the CNN convolutional operation in distinguishing MCI patients. In the binary classification task, the average accuracy of FCNN is 57.50%, with an MCI classification accuracy of 51.94%. In the triple classification task, the average accuracy of FCNN is 67.69%, with a classification accuracy of 100% for young individuals and 47.09% for MCI. This suggests that even though the eye-tracking and behavioral features we input are scalar, CNN or the convolution operations within it still play an important role in distinguishing MCI patients. FCNN does not exhibit similar classification ability.

Comments 7: Exploring other approaches, such as Support Vector Machines (SVM) and Random Forest, would strengthen the manuscript’s claims about the superiority of CNN in this context.

Response 7: We have added a comparison of the classification capabilities of different models in the Results section. First, regarding SVM, it shows weaker ability in distinguishing MCI patients from the control group of elderly individuals, with an average accuracy of 53.00%. The accuracy for distinguishing MCI is only 51.15%, which is not much different from random chance (50%). The classification performance of Random Forest is similar to that of SVM. In the binary classification task, Random Forest's average accuracy is 53.00%, with an accuracy of 47.80% for distinguishing MCI. In the triple classification task, Random Forest's average accuracy is 63.46%, with an accuracy of 100% for distinguishing young individuals, but only 43.6% for distinguishing MCI patients. Both SVM and Random Forest significantly underperform compared to CNN, which achieves an average accuracy of 66.50% in binary classification and 74.62% in triple classification. This result indicates the superiority of CNN in distinguishing MCI patients. For details, see Section 3.5.

Round 2

Reviewer 2 Report

Comments and Suggestions for Authors

No comments. The authors have answered all comments and questions in detail.